# Exploring the Potential of a School-Based Online Health and Wellbeing Screening Tool: Young People’s Perspectives

**DOI:** 10.3390/ijerph19074062

**Published:** 2022-03-29

**Authors:** Nicholas Woodrow, Hannah Fairbrother, Katrina D’Apice, Katie Breheny, Patricia Albers, Clare Mills, Sarah Tebbett, Rona Campbell, Frank De Vocht

**Affiliations:** 1School of Health and Related Research (ScHARR), University of Sheffield, Regent Court, Sheffield S1 4DA, UK; 2Health Sciences School, University of Sheffield, 3a Clarkehouse Road, Sheffield S10 2HQ, UK; h.fairbrother@sheffield.ac.uk; 3Population Health Sciences, Bristol Medical School, University of Bristol, Canynge Hall, 39 Whatley Road, Bristol BS8 2PS, UK; katrina.dapice@bristol.ac.uk (K.D.); katie.breheny@bristol.ac.uk (K.B.); patricia.albers@bristol.ac.uk (P.A.); rona.campbell@bristol.ac.uk (R.C.); frank.devocht@bristol.ac.uk (F.D.V.); 4Public Health, Floor 4, Halford Wing, City Hall, 115 Charles Street, Leicester City Council, Leicester LE1 1FZ, UK; clare.mills@leicester.gov.uk; 5Leicestershire Partnership NHS Trust, Bridge Park Plaza, Bridge Park Road, Thurmaston, Leicester LE4 8PQ, UK; sarah.tebbett@leicspart.nhs.uk

**Keywords:** schools, screening, child, adolescent, health, mental health

## Abstract

Despite high levels of need, many young people who experience health issues do not seek, access or receive support. Between May and November 2021, using semi-structured interviews, we explored the perspectives of 51 young people (aged 13–14) from two schools who had taken part in a novel online health and wellbeing screening programme, the Digital Health Contact (DHC). One school delivered the DHC during home-learning due to COVID-19 restrictions, whilst the other delivered it in school when restrictions were lifted. The DHC was seen as a useful approach for identifying health need and providing support, and had high levels of acceptability. Young people appreciated the online format of the DHC screening questionnaire and thought this facilitated more honest responses than a face-to-face approach might generate. Completion at home, compared to school-based completion, was perceived as more private and less time-pressured, which young people thought facilitated more honest and detailed responses. Young people’s understanding of the screening process (including professional service involvement and confidentiality) influenced engagement and responses. Overall, our findings afford important insights around young people’s perspectives of participating in screening programmes, and highlight key considerations for the development and delivery of health screening approaches in (and out of) school.

## 1. Introduction

Identifying and supporting young people’s mental and physical health needs is a recognised global public health priority [1]. Evidence suggests that many health practices, conditions and outcomes (e.g., smoking, mental health issues) become established during adolescence [2]. Whilst the prevalence of some adverse health practices seems to be reducing [3], worryingly, rising obesity rates [4], declining levels of wellbeing [5], and increasing rates of mental health problems among young people have been noted over the past decade [6,7]. A recent study suggested that internationally over 13% of young people have a mental health disorder [8]. This reflects rates in the UK, where it has been estimated that around one in six (16%) young people aged 5–16 experience mental health challenges, such as anxiety and emotional, behavioural or concentration issues [6]. Importantly, however, many more young people are suggested to experience adverse symptoms and effects of mental health conditions without meeting the threshold for clinical diagnosis [9]. More recent crises, most notably the COVID-19 pandemic, have intersected with ongoing system-wide challenges (e.g., deficits in specialist young people’s mental health provision [10]), and exacerbated health and wellbeing issues for many young people [11,12,13].

Despite this scale of need, it is well established that many young people who experience health issues do not seek, access or receive required support [14,15,16]. Young people with mental health needs have been noted to have particularly low levels of help-seeking behaviours [17]. This is especially concerning in light of the associated short- and long-term impacts of adverse health practices and conditions upon educational attainment, employment outcomes, and future health [18,19]. Crucially, access to support and early intervention are consistently associated with better health outcomes for young people [20,21,22,23]. However, there are a myriad of barriers to support-seeking and service engagement for young people [24]. Indeed, poor service availability and accessibility, inflexible treatment provision, and lengthy waiting times [16,25] have been noted as silent barriers. Overlapping with such system-level issues are a lack of knowledge about services [24], a lack of confidence in service and professional confidentiality [26,27], perceived stigma and embarrassment around service engagement [24,28], and a preference for alternative (informal) support avenues [27,28,29]. Facilitators to support-seeking and engagement are comparatively under-researched in contrast to barriers. However, there is evidence that young people perceive positive past experiences and knowledge, as well as social support and encouragement from others, as encouraging help-seeking [26,28].

### 1.1. Health Screening for Young People

There has been a growing emphasis on detecting health and wellbeing needs for young people, and the importance of the role of schools in this [30]. Schools provide an opportune context for early identification, due to their near universal contact with young people [31]. There are various methods of school-based screening, such as: curriculum-based models raising young people’s awareness of health issues and ways to address them; teaching school staff to identify and nominate ‘at risk’ young people; and the monitoring of risk markers (e.g., attendance levels) [21]. However, effective identification in the school setting is noted to be challenging [32,33,34,35]. Indeed, school staff have reported struggling to effectively identify young people experiencing issues, especially those with internalising issues [21,35,36].

There is growing evidence that universal school-based health screening offers the potential to help identify and provide support for young people with unmet health needs [21,31,34,37,38,39]. There is also growing evidence of the positive perspectives of school-based screening for health needs from parents and professionals [33,40,41,42], but less exploration of young people’s perspectives of their acceptability. Despite concerns around potential adverse effects from labelling [33], screening surveys have been suggested to be an effective and accepted method for exploring health issues [31], and an approach which potentially removes stigma and judgement, resulting in increased disclosure of need [20,43,44,45]. However, only a small number of schools utilise such tools [43].

Our study evaluates a novel school-based health and wellbeing screening programme which has linked follow-up support—the Digital Health Contact (DHC). Below, we provide a brief description of the DHC programme. A more detailed description can be found here [41].

### 1.2. About the Digital Health Contact (DHC)

The DHC is commissioned by Leicester, Leicestershire and Rutland councils as a non-mandated part of the 0–19 Healthy Child Programme (HCP), with Leicestershire Partnership NHS Trust (LPT) as the provider.

The DHC is an online health and wellbeing questionnaire delivered to an entire school year group (currently running in Year 7 (aged 11–12), Year 9 (aged 13–14) and Year 11 (aged 15–16)). The questions cover a range of physical and mental health topics, and have an option for providing additional information for each answer. The DHC acts as a universal screening tool, with indicated face-to-face support and follow-up from Public Health School Nurses (PHSN) for young people who are identified as having unmet health needs.

Following acquiring parental and young people’s consent, school teachers facilitate the questionnaire using school computers during lesson time (or using their own computer at home during COVID-19 lockdown periods; see below). All young people are provided with a digital personalised care plan upon completion of the questionnaire, which contains generic public health advice and signposting to relevant support. Specific responses or words/phrases from young people’s answers result in a ‘red flag’ referral being sent to the PHSN team. School staff members do not have access to young people’s responses, and do not see the ‘red flag’ referrals unless the PHSN team determines a safeguarding risk. Referrals are triaged by a PHSN who arranges a face-to-face appointment with the young person deemed to have an unmet need. During these appointments, the PHSN can deliver a range of support such as advice, signposting to other services or digital resources, an urgent referral to a specialist service (children and adolescents mental health services (CAMHS), Social Care), or further appointments and packages of work with the PHSN team. Because of disruptions and school closures due to the COVID-19 pandemic, in 2021, the DHC delivery method was altered, and young people completed the online questionnaire at home rather than at school. Following this deviation in delivery method, young people with ‘red flag’ referrals were offered an online video call appointment with a PHSN, or if deemed high risk, a face-to-face appointment.

### 1.3. Research Aims

Informed by a realist evaluation framework [46], this paper explores the perspectives and experiences of young people who have taken part in the DHC programme, including those identified as having an unmet need and offered follow-up intervention. Our key research questions were: (1) What are young people’s perspectives and experiences of participation in the DHC programme? (2) What are the strengths and weaknesses of the DHC programme, and what learning and recommendations can young people’s experiences provide for the delivery of the DHC? We have previously reported on perceived levels of effectiveness and feasibility of the DHC among programme stakeholders (providers and commissioners, PHSN and practitioners delivering the DHC, school leaders) [41].

## 2. Materials and Methods

### 2.1. Sample and Recruitment

We carried out interviews with young people who had participated in the DHC survey during the 2020/2021 UK academic year; we interviewed 51 young people in year 9 (aged 13–14) from two schools. Our sample included 32 female and 19 male participants; 47 participants were White-British (see Table 1 for an overview of the participating sample). Ethical approval for the study was granted by the School of Health and Related Research (ScHARR) ethics committee at the University of Sheffield.

While we initially sought to work with a diverse range of schools (in terms of urban/rural location, affluence/deprivation, ethnic diversity faith schools and single-sex schools), data generation during the COVID-19 proved challenging. Many schools moved to remote learning and did not participate in the DHC. When schools started reopening, fewer schools participated in the DHC or had the capacity to participate in the evaluation, resulting in only two rural schools being able to participate. Around 17% of pupils attending School 1 and 11% of pupils attending School 2 were in receipt of free school meals (we do not provide exact figures to ensure anonymity of our participating schools). This compares with a national average of 19.3% [47]. Approximately 7% of pupils at School 1 and 4% of pupils at School 2 did not have English as a first language. This compares with a national average of 16.9% [48].

While we aimed to recruit schools purposively, we were forced to take a pragmatic approach to school sampling due to the pandemic. However, we were able to purposively sample young people within each school. To gain insights into the perspectives of young people who followed different pathways in the DHC, the provider organisation selected a sample of young people from each school who had participated in the DHC. This ensured representation of young people who had received a ‘red flag’ and who were seen by a PHSN after initial triage (*n* = 15); young people who had received a ‘red flag’ and who were not seen by a PHSN after initial triage (*n* = 7); and young people who did not receive a ‘red flag’ (*n* = 7). Attempts were also made to ensure representation across gender, ethnicity, and disability, but this was dependent upon each school’s demographics. The selected young people were provided with a verbal description of the research project by school staff members, and asked if they wished to participate in an interview. School staff highlighted that the interview was to explore their perspectives of the DHC and not their specific responses to any questions. Those who said that they were interested in participating were given an information sheet (see Appendix A) and a consent form for them to read. We approached further young people with the same DHC status (e.g., young people who had received a ‘red flag’ and who were seen by a PHSN after initial triage) as those that said that they did not wish to participate. As young people were initially asked by school staff members if they would like to participate, their first expression of interest was not to a researcher or someone involved in the research, but to a familiar professional. We feel this detachment and gatekeeping from the school limited the impact of social desirability influencing the young people’s decisions, as they were not asked by, and thus did not need to respond to, a member of the research team about this. The process of gaining parental opt-in consent afforded young people more time to look at the study information sheet, contact the researchers/parents/teachers to discuss participation, and to decide whether to participate. After receiving parental consent, the young people were then asked by school staff members if they still wanted to participate. Those consenting completed a consent form, which was completed and returned electronically to the research team. In this way, young people were given detailed information around their participating rights, including the right to withdraw from the study, time to consider their participation and many opportunities to withdraw their consent from participation (including at the start of the interview when the information sheet and consent form were discussed and during the interview itself). All young people/parents consented from School 1, resulting in 29 participants. Six parents from School 2 did not consent for their child to participate, and one young person from School 2 decided, when asked at the start of the interview, that they no longer wanted to take part, resulting in 22 participants (see Table 1 for sample information). In this way, our approach blended both a purposive strategy and a pragmatic approach, reflecting the practical challenges we encountered [49].

Pupils from School 1 had participated in the DHC during home-learning when schools were closed due to COVID-19, whilst pupils from School 2 had completed the survey in school when they had reopened. Interviews took place between May 2021 and November 2021. Interviews were undertaken within one month of the young people completing the DHC screening questionnaire. Quotes from participants are presented including their reported gender (Female (F)/Male(M)) and their school and participation number (e.g., School 1 as ‘(M 1.1)’, and School 2 as ‘(F 2.3)’).

### 2.2. Data Generation

Interviews were arranged during school time and conducted through an online video conferencing platform in a private school office. All interviews were facilitated by author Nicholas Woodrow (NW). Interviews lasted between 25 and 30 min to ensure we did not take up too much of young people’s lesson time. The interviews followed a semi-structured topic guide (Appendix A) designed in consultation with young people in Patient and Public Involvement sessions. All interviews were audio-recorded using an encrypted recorder. The interviews were transcribed verbatim by a third party transcription company, anonymised at the point of transcription, and checked by NW for accuracy. To protect participant confidentiality, the research team, and researcher undertaking the interviews (NW), were not aware of potential participants’ status in the DHC (i.e., if they had been ‘red flagged’ or not), and it was made clear that the interview was not to discuss participants’ programme status or questionnaire responses, but to explore their experiences and perspective of the DHC.

### 2.3. Data Analysis

Interview data were analysed drawing on a thematic analysis approach [50]. We employed both inductive analysis (based on close reading of our transcripts) and deductive analysis (based on the topic guides and the aims and research questions of the project which NW used to develop an initial coding framework (Appendix A)). This approach ensured that we were able to answer the specific questions set out at the start of our project but also enabled us to incorporate aspects that we had not anticipated at the outset (e.g., ‘Online/face-to-face preference’ regarding the delivery of the DHC). All transcripts were coded by NW, and a selection of transcripts were separately coded by authors Clare Mills (CM) (*n* = 3) and Hannah Fairbrother (HF) (*n* = 5). NW then met with CM and HF to check for accuracy and consistency across coding. We used the data management system NVivo-12 (QSR International, Melbourne, Australia) to organise our coding.

## 3. Results

We identified two key themes (with sub-themes) from the analysis, ‘Perceived acceptability of the DHC’ and ‘Utility of support provided through the DHC’.

### 3.1. Perceived Acceptability of the DHC

Young people saw the DHC as a mental health and wellbeing (and less frequently, a physical health) ‘check-in’ tool, and a way to provide support and information for those not in contact with services. They generally reported the DHC as a quick, easy to use and helpful tool for discussing health. Young people thought the topics covered were pertinent and talked positively about the number, scope and range of questions. In terms of question wording, they generally described questions as easy to understand and interpret. The ability to expand on ‘closed’ answers through written responses was seen as an important feature, providing an opportunity to give as much or as little information as desired, and to clarify answers perceived as more nuanced than a ‘closed’ response could provide. Overall, they generally perceived the DHC as a valuable way of enabling young people to talk about their health and wellbeing, ask for support and, ultimately, receive support if needed:

‘*I do overall think it is a good way of doing it as it encourages people to talk about it and to talk about how they*’*re feeling*.’ (F 2.7)

Young people highlighted that the DHC raised awareness of, and helped them to reflect on, issues that they perceived to have been ‘normalised’, and which might require support.

‘*Well*, *I mean*, *it kind of asks the questions that you don*’*t really want to ask yourself*, *the ones that you probably wouldn*’*t be made even aware of that are a problem*, *like*, *without being asked it*, *like certain ones that you will think oh that*’*s normal*, *that*’*s not anything wrong and then it*’*s on the survey like oh that*’*s actually something that shouldn*’*t be happening, maybe I should be talking to someone about it*.’ (M 1.3)

They also foregrounded the contextual and signposting information around each question, which they perceived as helpful. Participants highlighted the provision of a tailored care plan of relevant information and support following questionnaire completion, providing opportunities for young people to ask for and receive help when they may not have previously:

‘*I do think it*’*s a good way to do it and I definitely appreciated it because then from that I started meeting the school nurses*, *so I think it*’*s a good way for students to get that first step to get help if they need it*…*I do think that it helps because a lot of students might be afraid to come forward or not really know who to go to and so this survey gives them like an opportunity to ask for help without really having to ask for help*.’ (F 1.22)

While young people did not envisage the DHC as overcoming all barriers around support-seeking for young people, they highlighted how it offered a beneficial, alternative proactive avenue for support. They described having different avenues for support-seeking as a way of helping to ensure different options to complement young people’s eclectic preferences:

‘*As many options as possible is the best thing*, *because everyone is going to want something different*, *not everyone is going to want online*, *face to face*…*you never know what each person is going to prefer*, *so I mean I think as many options as you can have to talk to someone about it is definitely the best thing you can have in that sense*.’ (M 1.3)

#### 3.1.1. Delivery Mode

##### Online Delivery

The online delivery of the DHC screening questionnaire (whether in school or at home) was a salient theme in the participants’ perception of its value. Indeed, it was seen as a comfortable space to articulate their concerns, and as an appropriate way to enable expression for those who struggle raising and talking about sensitive and personal issues with others face-to-face:

‘*I think you could be more honest online*. *Sometimes if you speak to someone you might not say everything you want to say*…*I think some people find it like*, *awkward and things to speak to people about it*.’ (F 2.13)

For many, the online delivery was seen as enabling more open and honest responses, and the ability to discuss topics around health without feelings of embarrassment (which face-to-face questioning was suggested to have potentially caused). This described how this approach afforded a perceived level of control to those participating, and suggested that it supported disclosures:

‘*It definitely helped me like tell someone about like something that I*’*m not too keen on sharing and the fact that I wasn*’*t doing it like face*-*to*-*face with someone*, *I think that really helped because I probably would have been really awkward and wouldn*’*t know what to say*.

*Interviewer*: *I mean*, *I*’*m not asking you to tell me about what you put but do you think you would have talked about that if this survey wouldn*’*t have been sent to you or was that something that* –

*I probably wouldn*’*t have said everything*. *I would have said most of it but I probably wouldn*’*t have said some of the serious things if I did*.’ (F 1.29)

Conversely, and highlighting the importance of various avenues of support, a small number of participants said that they would have preferred a face-to-face rather than online approach. They thought face-to-face would be more personal, aid clarification, and facilitate more open responses (due to it being harder to hide issues if asked by a PHSN directly).

‘*with me I find it easier to talk to people in person than do it online*. *I feel like when I*’*m with somebody and I feel like I can trust them*, *I*’*ll just open up about everything*, *but might be different for some other people*.’ (F 1.19)

##### Home/School-Based Delivery and Completion

As noted above, our sample included one school whose pupils completed the DHC during home learning, and one whose pupils undertook the DHC in school. This enabled an interesting juxtaposition of perspectives around programme delivery. The majority of participants reported feeling comfortable completing the DHC screening questionnaire irrespective of the context of delivery. Many participants (typically those young people who did not receive a ‘red flag’, and thus did not disclose any issues during the questionnaire) noted that completion in school around other young people did not alter their responses. However, highlighting the impact of delivery context and perceived privacy upon responses, some young people reported that completion in school did impact their responses, due to the presence of both peers and teachers. They described how this altered some of their responses and the amount of detail they provided for follow-up questions:

‘*There were one or two that I kind of just put a brief explanation because*, *like I said*, *I just kind of wanted to get it out the way so people wouldn*’*t see*...*I think I*’*d probably put a bit more detail in it if it was at home*, *just because also when we*’*re at school*, *obviously like I said*, *there are people that could look over and see*.’ (F 2.21)

In particular, young people described how the layout of the classroom was unconducive to privacy: ‘because our computers are, you’ve got people back to back and felt like people were looking at answers and I didn’t want anyone to say something that someone saw’ (F 2.8). Indeed, the structure of the DHC is such that if a young person gives a response suggesting a potential health or wellbeing need, they are invited to type in a more detailed answer. This means that typing inevitably signposts ‘a problem’ to those around the young person. Whilst the majority of participants noted they answered honestly and would not have altered their answer irrespective of the delivery context of the questionnaire, there was a general perception that home delivery (or delivery in school in a more private way) was preferred: ‘[It’s] better at home because then you don’t feel like anyone’s judging you around the class. I probably wouldn’t have answered so honestly if I were to do it inside of school.’ (M 1.12)

Conversely, a minority of participants reported a preference for delivery in schools. They thought that a teacher or school staff member would be able to offer help in understanding any tricky questions, overcoming any technical problems, and providing any direct support. There was also a consideration of how homelife issues (problematic relationships with parents) may have inhibited honest responses for some if completing at home. Nevertheless, the ability to complete the DHC privately and independently at home was generally seen to add a level of protection around privacy and ‘safety’ which was difficult to achieve in schools. Young people thought that ensuring privacy would help them to provide more detailed and honest responses. They also described how completion at home afforded more time for the questionnaire. Indeed, some young people who completed the questionnaire in school noted having limited time: ‘[we had] about five to ten minutes but teachers would start rushing after a while, we were told to log off and stuff’ (F 2.20). Having more time to complete the questionnaire was suggested to improve reported honesty, and reduce issues in comprehension:

‘[*at home*] *you can have a think about it and you don*’*t have to rush through it thinking that you don*’*t have enough time*…*you can just sit there at your desk or on your bed or something and you can think about questions and you can answer them truthfully and you*’*re not limited to a time*.’ (M 2.22)

#### 3.1.2. Understanding of the DHC Screening Process

As well as delivery context, the participant’s understanding of the screening process (why they were completing it, where their responses went and who saw them, what happened following this) were noted as important components of engagement. Indeed, young people highlighted how their level of understanding around the DHC process, based on how it was presented and explained to them, could impact upon the detail and honesty of responses. There was variation in how the participants recalled being presented information (from a presentation about the DHC, to receiving information sheets and verbal descriptions from their teachers). Whilst there is a description at the start of the DHC screening questionnaire regarding its process, young people talked about how they did not always understand or retain this information. The importance of young people having a good understanding of the DHC, and information being presented in a consistent way, was evident. For some participants, there was confusion and concern around where their responses would be sent, and who would be able to access them. The explanations given by school staff members around key aspects such as confidentiality were not described as consistent. For example, some young people noted being told that the DHC screening questionnaire was ‘completely confidential’, and were surprised to find this not to be the case: ‘It’s a bit annoying, because they said that it’s all confidential, but then it red flags it and sends it to them, and then they have to have someone coming in to speak to us. It’s a bit annoying.’ (M 2.14)

There was a perception, primarily from those who had a less clear understanding of the DHC process, that school staff members would have access to responses. School staff were perceived by the participants as more likely to be judgmental, to ‘overreact’, and to share information with others without consent. Indeed, a barrier to openness for the young people (including the level of detail in responses) was concern around who would be able to see responses. Uncertainty around DHC processes was noted to impact honesty and reticence in responses:

‘*There was obviously that uncertainty of where it was going and I think some forms got told where it was going but I know our form didn*’*t really get told where it was going so there was some uncertainty of you know how should I answer*.’ (F 1.27)

‘*You don*’*t want to admit something if you don*’*t know where it*’*s going because you*’*re like* “*Well anyone could see this*” *whereas if you know it might have helped some people*.’ (M 2.6)

Some participants spoke of the importance of informing young people of the DHC process and potential outcomes, despite this potentially impacting upon honesty in responses:

‘*I think it could go one of two ways*. *I think it could either make people not want to put things because maybe they don*’*t want to be talked to or don*’*t want to talk about it*, *or it might encourage people to put it because they do want to talk about it*...*I think more openly because I know like some people*, *like my friends*, *when they found out that like people who answered like and it was concerning and they were getting help*, *were like oh I wish I*’*d done that because I lied and I think I actually need to talk to somebody*.’ (F 2.20)

Overall, the delivery context of the DHC and understandings of the screening and follow-up process were described as shaping responses and perceptions of the acceptability of the DHC programme.

### 3.2. Utility of Support Provided through the DHC

#### 3.2.1. Increasing Knowledge and Use of Support Options

All participants in the DHC received health promotion information during and at the end of the screening questionnaire. Not all participants could recall the specific information they received from the questionnaire and PHSN appointments. Those who did, however, reported that the signposting information and personalised care plans provided useful and relevant health and wellbeing information and support, advice and coping strategies, and signposted previously unknown avenues of support. This included support within schools, ‘[the DHC] opened up about quite a few places within the school that I could get help from that I didn’t know about before’ (F 1.11), and broader avenues of support:

‘*I think before I knew about the quiz thing*, *there wasn*’*t a lot of support for mental health*, *there wasn*’*t a lot of advertising for it really*...*but now I*’*ve been to the appointment*, *I got given like a list of a few websites I can look at and stuff*.’ (F 2.21)

They described how participation in the DHC had helped to raise their awareness and knowledge of PHSN services: ‘I had no clue that we had school nurses before, really. I didn’t know we could actually talk to people other than the teachers’ (F 1.23). It also highlighted PHSN roles in relation to support for physical and mental health problems:

‘*Yes I heard of* [*PHSN*] *before but I thought it was literally just if you hurt yourself*, *you could go. I didn*’*t really get told it was for like mental health and stuff*...*because I*’*ve been to see the school nurses* [*through the DHC*], *I know like they*’*re different now but before that literally I would have just thought you go there if you have a stomach bug*.’ (F 2.20)

Young people were particularly keen to highlight their preference for reaching out to PHSN rather than school staff as they perceived this would afford a higher degree of confidentiality:

‘*well the teachers talk to each other don*’*t they*, *but as a school nurse it*’*s their job to do something like that*, *so I think they would understand why you want to keep it confidential and not talk to anyone else about it*.’ (M 2.3)

Though there was evidence that the DHC had increased young people’s awareness of the PHSN role, this typically came through in interviews with young people who had received some form of PHSN contact following the screening questionnaire. Even amongst this group, there was still some uncertainty regarding how to directly contact PHSN: ‘The survey made me aware that there are people that can help you in school if you’re feeling down. I don’t know how to contact them. You’d probably like go to reception, ask for somebody’ (M 2.15). Participants suggested that advertising the role of PHSN and how to contact them both during and following the DHC screening questionnaire would be helpful.

#### 3.2.2. PHSN Follow-Up Session Support

The system of appointments being made by the PHSN was generally seen as a beneficial aspect of the DHC programme. Participants described how this reduced personal effort on young people’s part, and helped to ensure that young people who may have not actively contacted the PHSN to arrange an appointment were still given an opportunity to be seen:

‘*I think it*’*s quite useful that the nurses do it instead of the students themselves because it might be more stressful to make an appointment for yourself and you might not necessarily want to because you might be too nervous or anything*.’ (F 2.10)

However, highlighting gaps in their understanding of the DHC process, some young people described feeling shocked and worried about being contacted after completing the questionnaire:

‘*It was a bit of a shock really*. *I didn*’*t know that it was happening*. *I just got called out from one of my lessons and we went into a private room and it was me and then there was two other nurses*, *so I was kind of a bit like shocked by it all*, *but then after that it*’*s just kind of*, *yeah*, *it’s not been that big of a problem at the minute*, *they*’*ve kind of dealt with it and*, *yeah*, *so they*’*re helping out now*.’ (F 2.21)

Some young people also suggested they would feel more comfortable if there was more communication with PHSN when appointments are made, and if they had active input in the times they were seen, rather than being passive recipients of appointment slots: ‘I think it’s good that the nurses do it but if there was a bit more communication with it, like the students could pick a certain time that was available for the nurses and them, it would help a lot more’ (M 1.25). They also highlighted that having notice and extra time would help them prepare themselves for the appointments: ‘I’d probably prefer them to ask me so that I know about it, so that I can like think more about it and what to say to them’ (F 1.8).

The participants who were involved in follow-up support with PHSN spoke positively about their experiences of care and support provided:

‘*Well for some of the questions that I put down*,* or answers sorry, they were a bit concerned about, which I did speak to the school nurse about and she just gave me some information on what to do*,* like with anxiety and mental health and stuff like that and it really did help*.’ (F 1.19)

A valued aspect of the follow-up sessions was that the PHSN had background information around the issues the participants were experiencing, with this importantly being in young people’s own words through the DHC screening questionnaire responses. The participants noted that this made them feel more comfortable as they did not need to start discussion themselves:

‘*Well if you*’*ve already done the thing and then they need to talk to you*, *I think it*’*s easier because you don*’*t need to explain what you*’*re anxious or whatever*, *because you*’*ve already put that and they already understand*...*I think it*’*s good because you don*’*t have to repeat yourself*, *but you can if you need to*, *that they already know all the stuff and you don*’*t have to say it if you don*’*t want to*.’ (F 1.10)

## 4. Discussion

This study has explored young people’s perspectives of a novel school-based online health and wellbeing programme—the DHC (see Table 2 for a Summary of key findings). Overall, the participants reported the DHC as a helpful and useful way to talk about their health and receive support for issues they are experiencing. Young people valued the online delivery of the questionnaire and described how this encouraged them to be more honest and open in their responses. The opportunity to complete the questionnaire at home was also perceived as beneficial, giving more privacy and more time for participants. In turn, young people perceived this as facilitating greater honesty and detailed responses. Beyond the delivery context, knowledge and understanding of the screening process (including its ‘separation’ from the school) was important for engagement and impacted upon young people’s reported quality, quantity and level of openness in completing the questionnaire. That the DHC is run, managed and arranged by PHSN (as opposed to school staff) was perceived to be a key facilitator of engagement. The DHC programme was seen to increase awareness of support options for young people with health and wellbeing needs, but knowledge of how to directly access support was limited. The DHC appears to be a beneficial way of identifying need, providing support and increasing awareness and willingness to engage with services, which may help overcome some barriers associated with help- and support-seeking in young people.

Health-seeking practices in young people are complex, but the benefit of early intervention for mitigating long-term health and wellbeing issues is evident [21,22,23]. Our findings highlight the importance of having a broad range of approaches for the identification of need, to ensure there are options which young people feel confident and comfortable using. Our study adds to the evidence of school-based screening surveys having a high level of acceptability and being perceived as a useful approach for exploring health and wellbeing issues with young people [39,51,52]. It highlights the potential of school-based universal health and wellbeing screening with linked follow-up intervention as supporting identification and providing support for young people with unmet health needs [20,21,23,31,33,37,38,39,45]. Screening models such as the DHC may afford a way of responding to recent calls in the UK to improve mental health support and treatment in schools [53], and investment to boost mental health support for children and young people to reduce current treatment gaps [54]. Indeed, such models may help address ‘systemic and structural’ barriers around support-seeking, helping overcome noted barriers of accessibility, time and transport costs [24]. Importantly, our work highlights how young people value interventions being ‘brought to them’ and the lack of individual effort in support-seeking this provides, but they do not just want to be passive recipients, and want some ownership around the process and their place in it (e.g., involvement in the organisation of appointments).

Our study shows how the online delivery of screening tools may aid a level of ‘detachment’ from directly speaking to a person, and foster the removal of perceived embarrassment and judgment associated with this. This is crucial as reducing perceived stigma from young people has been noted as important in encouraging the disclosure of sensitive information [20,44,45] and aiding mental health support-seeking [24,55]. Indeed, the online delivery of the DHC screening questionnaire facilitating disclosures from young people supports previous research which highlights the benefits of digital approaches around encouraging self-referral from young people [56].

Our findings also highlight how the delivery context of screening tools can have important impacts upon engagement. Whilst schools may be a ‘safe’ context for many young people, for some, they may inhibit disclosures (through reduced honesty and openness in responses) due to perceptions of privacy [28,57,58]. Concerns around privacy being effectively managed, particularly by school staff [24,25,27], echo findings from the wider literature [59,60]. Worries around school staff insensitively handling disclosures and parents finding out private information have been noted as barriers impacting upon young people’s willingness to seek support from school-based mental health services (see [61]). Our findings support this and highlight how such concerns can impact upon screening responses.

Whilst we found young people to have a general preference for home over in-school completion of the screening questionnaire, it is important to highlight the potential for a move to home completion to widen inequality. Not all young people have access to the internet and technologies at home [62,63] or a private space in which to complete screening questionnaires. This means the most deprived and disadvantaged may be excluded. Delivery in schools crucially provides a universal and accessible option for young people, potentially ameliorating impacts of disadvantage upon participation.

Our study highlights that whilst there are often many support options and resources available for young people, a lack of awareness of available services can impact upon help-seeking practices, which echoes previous findings [14,60,64]. The DHC acted to increase both knowledge of and access to resources and support. However, importantly, our findings underscore that knowledge around how to directly access and contact services, and knowledge around service processes and confidentiality, play crucial roles in facilitating support-seeking [24,65]. Knowledge around the process of screening programmes (specifically which services/professionals are involved) appears salient in facilitating engagement and reported honesty. In our study, the ‘separation’ between the school and the PHSN in the running of the DHC was an extremely valued aspect of the programme, but was not well understood by all young people. The importance of highlighting such separations and the process of interventions was clear, with this reported to have an impact upon the depth, detail and validity of responses.

### 4.1. Practice Implications

There is mixed evidence on the feasibility, effectiveness and cost effectiveness of screening approaches [20,21]. Our study affords important implications from the perspectives of young people.

To facilitate participant engagement, honesty and detail in screening responses from young people, it is crucial to clearly highlight programme processes (e.g., who has access to participant responses, the protections and limits of confidentiality). Consistent messaging around this from those presenting and delivering screening programmes is also important.Whilst knowledge of support options can be increased through participation in screening with linked follow-up programmes, the importance of further advertising, promoting and reinforcing how to directly contact support (e.g., PHSN) is vital.Ensuring privacy and adequate time to complete screening questionnaires (through flexibility of delivery context, e.g., at home, or through increased privacy in school settings) may encourage honesty and detail in responses.Greater process knowledge and improving young people’s involvement in the setting up of appointments may help reduce worry and anxiety around linked follow-up sessions. Sensitive handling of follow-ups may also reduce issues [66].

### 4.2. Study Strengths and Limitations

This is the first study to focus specifically on young people’s perspectives of the DHC, and the acceptability of the DHC and its abilities to identify and provide support for unmet physical and mental health needs, building on previous work exploring the perspectives of key stakeholders involved in the DHC programme [41,67]. In light of challenges in screening programme feasibility [20], and limitations around implementation of the DHC programme (including challenges around securing school engagement/participation [41]), exploring young people’s perceptions of acceptability provides beneficial insights into their experiences of school-based screening programmes.

Our sample was a small purposive sample drawn only from two schools. Due to the ongoing challenges from the COVID-19 pandemic, we experienced challenges in school recruitment. We had planned and attempted to recruit up to four schools, reflecting as diverse a sample as possible (rural, urban, single-sex school, religious schools). However, schools moving to online learning resulted in fewer schools participating in the DHC, leaving less to recruit from. School COVID ‘bubbles’ (class or year groupings implemented to reduce COVID transmission) having to self-isolate also complicated data generation. As a result of the pandemic, we had to take a pragmatic approach to the recruitment of schools. We were only able to capture the perspectives of young people from two schools similar in location and demographics. Further, while we attempted to draw a sample of young people who reflected a diverse gender and ethnicity mix, our sample demographics were dependent upon the school year demographics, and which young people consented to participate. Our sample was mostly female (63%) and white-British (92%). A wider mixed gender and ethnicity sample would have allowed intersections to be further explored, as would participation of a more diverse selection of schools (e.g., urban location, single-sex school, faith schools). Further, since both schools had lower than the national average levels of free school meal eligibility, our study affords limited insight into how perspectives may differ between young people from contrasting socioeconomic backgrounds. There are acknowledged limitations of purposive sampling which impact the generalisability of findings [68]. We appreciate limitations in our sampling approach which means we have potentially missed insights and perspectives of the DHC; this would benefit from further research. Indeed, we initially wanted to explore a more diverse range of young people and schools, looking at a range of urban and rural locations, ethnicities, and different contexts of affluence and deprivation. Nevertheless, despite such limitations, our findings offer important insights from young people around the delivery of public health interventions in schools.

An important consideration, and arguably a strength of this research, is that the young people from our participating school participated in the DHC screening questionnaire when it was run both at school and home. This provided a useful comparison between the delivery context of the survey. Due to this, our findings importantly show the potential impact of this context upon reported engagement and honesty. However, due to our limited sample, factors such as classroom layouts may have been different in different schools, impacting upon the findings. Nevertheless, perceptions from participants across both school samples highlighted concerns around privacy from teachers and peers if completed in school.

### 4.3. Implications for Future Research

There are limited studies looking at young people’s perspectives of online school-based screening programmes and screening tools more broadly [31,33]. Thus, to better understand the acceptability of school screening tools, and to better gain insights around future development, research must engage with a broad range of young people.

As we were unable to work with a variety of schools over different locations and contexts (e.g., of urban/rural and affluence/deprivation) due to recruitment challenges during the COVID pandemic, we have identified a number of priorities for future research. First, future research should explore the perspectives of young people living in more deprived contexts. In particular, as alluded to in the discussion, it should explore whether the preference for home-based versus school completion stands for this demographic. It would also be beneficial to explore the perspectives of young people from black and minority ethnic groups and young people attending faith schools. There is evidence to suggest that there can be cultural factors which act as barriers/facilitators to support-seeking [14,69]. Exploring different groups’ perspectives of the acceptability of the DHC and online screening tools more generally may have important implications for screening programmes.

Further, while there is some work looking at parents’ perspectives of school-based health interventions and screening programmes [40,70], due to COVID-related delays in setting up the study, we were unable to recruit and explore parents’ perspectives in our project. It is important to look at parents’ perspectives of acceptability of screening programmes to help facilitate their implementation. Indeed, as parental consent is needed for participating in the DHC, the withdrawal of consent may result in young people with health needs being missed. Any further evaluation of the DHC and its wider roll-out, therefore, should seek to explore parents’ perspectives too.

## 5. Conclusions

The DHC was seen as a useful and beneficial approach for identifying health need and providing support for young people, and one that had a high level of acceptability from participants. The context of screening delivery (online, home/school-based) was noted as having an impact upon perceived privacy and reported honesty in responses. The importance of young people having a clear and robust understanding of the process of an intervention was important in ensuring detailed and honest responses, and in facilitating effective engagement. Overall, our study highlights the potential utility of screening programmes with identified follow-up support from PHSN. Our findings further provide important insights into the perspectives of young people participating in screening programmes, and highlight useful considerations which may be beneficial in the development and delivery of health and wellbeing screening approaches in (and out) of school. Our study underscores the value of exploring the perspectives and experiences of young people involved in health interventions, in order to better inform and shape the delivery and practice of public health work.

## Figures and Tables

**Table 1 ijerph-19-04062-t001:** Sample overview.

	School 1 (DHC Delivered at Home)	School 2 (DHC Delivered in School)	Final Sample
Sample information	29 participants	22 participants	51 young people
	21 female 8 male	11 female 11 male	32 female 19 male
	25 White-British 6 Asian/Asian British	22 White-British	47 White-British 6 Asian/Asian British
Sample status in the Digital Health Contact (DHC) programme	15 had received a ‘red flag’ and were seen by a Public Health School Nurse (PHSN) after initial triage	11 had received a ‘red flag’ and were seen by a PHSN after initial triage	26 had received a ‘red flag’ and were seen by a PHSN after initial triage
	7 had received a red flag and were not seen by a PHSN after initial triage	6 had received a red flag and were not seen by a PHSN after initial triage (*n* = 7)	13 had received a red flag and were not seen by a PHSN after initial triage (*n* = 7)
	7 did not receive a ‘red flag’	5 did not receive a ‘red flag’	12 did not receive a ‘red flag’

**Table 2 ijerph-19-04062-t002:** Summary of key findings.

Theme and Sub Themes	Key Findings
Perceived acceptability of the (Digital Health Contact) DHC - Delivery context (online, home/school-based) - Understanding of DHC screening process	The DHC was described as a useful way to talk about and receive support for health issues they are experiencing. The online delivery of the DHC screening questionnaire was seen to encourage participants to be more honest and open in their responses.
	Completing the questionnaire at home, compared to in school, was seen to give more privacy and more time for participants. This was noted as facilitating honesty and detail in responses. However, not all young people have a private space or access to the required technologies at home. Thus, delivery in schools crucially provides a universal and accessible approach for young people. Knowledge and understanding of the screening process (including its ‘separation’ from the school; who sees responses; what can happen following compilation of the questionnaire) impacted upon young people’s reported engagement, and quality, quantity and level of openness in responses.
Utility of support provided through the DHC - Increasing knowledge and use of support options - PHSN follow-up session support	The DHC programme was seen to increase awareness of support options for young people with health and wellbeing needs (both in and out of school), but knowledge of how to directly access support was limited. The managing of the DHC by Public Health School Nurses (PHSN) (as opposed to school staff) was valued, and perceived to be a key facilitator of engagement. The system of appointments being made by the PHSN was generally seen as a beneficial aspect of the DHC programme, one which reduced effort on young people’s part.

## Data Availability

Data available on request due to restrictions. The data presented in this study are available on reasonable request from the corresponding author. The data are not publicly available due to privacy reasons.

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
