# Peer review of "Exploring the Potential of a School-Based Online Health and Wellbeing Screening Tool: Young People’s Perspectives"

_ijerph, 2022, doi:10.3390/ijerph19074062_

Round 1

Reviewer 1 Report

The study qualitatively examines the online health and wellness screening programme - Digital Health Contact (DHC) among two schools in the UK. 

The abstract is not structured. Did not follow the IJERPH format. 

Firstly, the authors did not follow the reference format by IJERPH. 

Please give more information about the Digital Health Contact (DHC) programme, as I googled and could not find anything about it. Please mention the difference between DHC programme published at BMC Public Health and this. 

Methods

Is this more toward convenient sampling rather than purposive sampling? The limitation of using purposive sampling should be further explained in the limitation section. 

I could not understand the importance of Table 1. Kindly explain this further. Table 2 seemed to be enough information. Thus, I suggest deleting Table 1. 

Suggest including a sample size calculation, conceptual framework and hypothesis testing. 

Who did the interview? How was the interview conducted? How long is the interview? Many missing information. Is the data reaching saturation? How to ensure that the data given by the young kids are correct? 

Even though this is a good study, I felt the authors are marketing the programme more than writing a constructive understanding. What is the limitation of the programme? Why did not many schools take up this programme? Who is involved in this programme? Many information is missing. 

I suggest the authors look into it and restructure the manuscript towards an academic manuscript. 

Again, references also did not follow the IJERPH format. Disappointing as it should be a priority for authors. 

All the best. 

Reviewer 2 Report

Dear authors, the study is of moderate interest and requires minor revisions, it is a novelty but it is a qualitative study and has all the limitations of this type of study. Sampling could be an important limitation. Here are some indications:

-Please enter the study period in the abstract.
-Please use MeshTerms as keywords whenever possible.
-Please explain the sampling better and report in depth in the discussions that it could be an important limitation of the study.
-Please describe the sample in more detail using the tables, a brief statistical analysis would also be appropriate to see if there are statistically significant differences.
-It would be advisable to make a summary table with the main results obtained, it would facilitate reading.

Please answer point by point.

Thank you.

Best regards.

Reviewer 3 Report

The manuscript is generally okay except for (1) the method section (2) a similar prior publication: https://doi.org/10.1016/S0140-6736(21)02634-9

And the problems mentioned above have enormous implications for the rest of the manuscript.

To illustrate, they negatively impact parts of the current submission and devalue them. The issues undermine different areas of the article, such as the results, discussions, and recommendations.

(A)

For example, on page 4, the authors attempt to tell readers:

  1. The selected young people (13-14 years) were asked if they would like to participate in an interview for this research. 
  2. Those who expressed an interest were provided with an information sheet and consent form. 
  3. Parental opt-in consent was also acquired. 
  4. After parental consent young people were again asked if they wished to participate. 
  5. Those consenting completed a consent form, which was completed and returned electronically. 
  6. All young people/parents consented from School 1 resulting in 29 participants. 
  7. Seven young people/parents did not consent from School 2, leaving 22 participants. (See Table 2 183 for sample information).

Based on the above pieces of information, I have the following questions for you as researchers:

  • Did you ask them AGAIN if they wished to participate after the fact, that is, (1) after 13-14-year-old children had verbally agreed & (2) after you had received written consent from their parents? 

  • Why and to what effect? 

  • What happened when you indeed asked them?

  • Could social desirability be the determinant factor in their agreement to participate?

  • How informed were they about their rights to withdraw from participating without repercussions (social, educational, cultural)?

  • You reported that "Seven young people/parents did not consent from School 2". Was this before or after you asked them for the 2nd time if they were willing to go ahead as participants for your study?

A more appropriate account is only possible if investigators follow the proper steps to collect data. 

Ultimately, it reinforces a critical point about research practices that the way in which knowledge is actually acquired is vital to what the claims are. 

(B)

On page 13, you claimed:

"This is the first study to explore young people's perspectives of the acceptability of the DHC....

The above claim is inaccurate because a similar article was published in 2021: https://doi.org/10.1016/S0140-6736(21)02634-9

For example, see:

  • Woodrow, Nicholas, Hannah Fairbrother, Katie Breheny, Kartina d'Apice, Patricia Albers, Clare Mills, Matthew Curtis et al. "Exploring the potential of a school-based online health and wellbeing screening tool: professional stakeholders and young people's perspectives and experiences." The Lancet 398 (2021): S91.

Round 2

Reviewer 3 Report

The paper has improved following the revision of some parts.

However, the authors' claim on page 15 is still incorrect:

"This is the first study to specifically focus on young people's perspectives of the DHC....

Citing the previous study to support the sentence cannot do. Why make an effort to conceal the achievements of prior publication output, one may ask?

The author could claim that this current submission is one of the few studies to "specifically focus on young people's perspectives of the DHC...." and then mentioned previous similar publications because the article below is a publication in its own right. One can rightfully cite the prior publication in place of the current submission as the first study. It is confusing and incorrect.

See:

  • Woodrow, Nicholas, Hannah Fairbrother, Katie Breheny, Kartina d'Apice, Patricia Albers, Clare Mills, Matthew Curtis et al. "Exploring the potential of a school-based online health and wellbeing screening tool: professional stakeholders and young people's perspectives and experiences." The Lancet 398 (2021): S9, https://doi.org/10.1016/S0140-6736(21)02634-9 

Author Response

Reviewer 3. Comment 1

The paper has improved following the revision of some parts.

However, the authors' claim on page 15 is still incorrect:

"This is the first study to specifically focus on young people's perspectives of the DHC...."

Citing the previous study to support the sentence cannot do. Why make an effort to conceal the achievements of prior publication output, one may ask?

The author could claim that this current submission is one of the few studies to "specifically focus on young people's perspectives of the DHC...." and then mentioned previous similar publications because the article below is a publication in its own right. One can rightfully cite the prior publication in place of the current submission as the first study. It is confusing and incorrect.

See: Woodrow, Nicholas, Hannah Fairbrother, Katie Breheny, Kartina d'Apice, Patricia Albers, Clare Mills, Matthew Curtis et al. "Exploring the potential of a school-based online health and wellbeing screening tool: professional stakeholders and young people's perspectives and experiences." The Lancet 398 (2021): S9, https://doi.org/10.1016/S0140-6736(21)02634-9

Response:

We feel that the reviewer is confusing our conference proceeding paper which is simply an abstract describing an oral presentation (Woodrow, Nicholas, Hannah Fairbrother, Katie Breheny, Kartina d'Apice, Patricia Albers, Clare Mills, Matthew Curtis et al. "Exploring the potential of a school-based online health and wellbeing screening tool: professional stakeholders and young people's perspectives and experiences." The Lancet 398 (2021): S9, https://doi.org/10.1016/S0140-6736(21)02634-9), with a published paper. As we noted in our previous response, this was a conference presentation where we orally presented initial emerging findings of our evaluation project more generally to help shape our academic papers.

We are not trying to conceal this and are happy to cite it. However, it is important to be clear that there are no published papers exploring young people's perspective of the DHC. We feel the request to reframe our study as 'one of the few studies to specifically focus on young people's perspectives of the DHC' as the reviewer suggests, would lead to confusion for any readers, who may search for other studies or published papers and only find our conference proceedings paper/abstract. 

We feel we present, discuss and cite our work appropriately